materials science/green chemistry

water-borne alkyd resin, waste polyethylene terephthalate, physical properties, hardness, water resistance

**Author for correspondence:**
Yongbo Ding
e-mail: yongboding@jxstnu.com.cn

# Preparation of autoxidative water-reducible alkyd resins from waste polyethylene terephthalate

Siming Ouyang, Yuqing Xie, Wangxing Fu, Yongbo Ding and Liang Shen

Department of Coatings and Polymeric Materials, College of Chemistry and Chemical Engineering, Jiangxi Science and Technology Normal University, Nanchang 330013, People's Republic of China

In this paper, the waste polyethylene terephthalate (PET) was glycolysed by trimethylolpropane with zinc acetate as catalyst. The effects of different content glycolysis product of waste PET on the appearance, viscosity, particle size and molecular weight of autoxidative water-reducible alkyd resins and the corresponding film adhesion, flexibility, impact resistance, gloss, hardness and chemical resistance were studied. Meanwhile, experimental results were compared with commercial water-reducible alkyd and water-reducible alkyd without the glycolysis product of waste PET. The results show that the maximum concentration of PET in autoxidative water-reducible alkyd resins can reach 8.5 wt%, and the molecular weight, particle size and viscosity of water-reducible alkyd resin do not change much with the increase of PET concentration. The introduction of PET resulted in the viscosity of water-reducible alkyd resins being greater than that of water-reducible alkyd resin without PET; this is mainly because PET contains harder terephthalic acid monomer units. However, the particle size of water-reducible alkyd resins with waste PET is significantly lower than that of the water-reducible alkyd resin without PET; this is due to PET-free water-reducible alkyd resin containing more pentaerythritol with greater steric hindrance. In addition, the hardness of the water-reducible alkyd resin paint film (PET content is 8.5%) reaches 1H, which is higher than the hardness (HB) of the water-reducible alkyd resin paint film without PET and the commercial alkyd resin paint film, while the physical properties and chemical resistance of the former are comparable to those of the latter two kinds of paint films. Therefore, the use of waste PET in water-borne coatings systems not only reduces the cost of coatings, but also opens up a new market for recycled PET, which may contribute a promising method for management of waste PET.

# 1. Introduction

Polyethylene terephthalate (PET) possesses excellent transparency, stability and plasticity [1], so it is widely used in food, medical treatment, industry and other fields [1,2]. For example, almost all mineral water and beverage bottles are made from PET, which led to waste PET bottles becoming the largest proportion of solid waste [1]. Waste PET is not easily decomposed in natural conditions, which wastes a lot of space and results in serious environmental pollution [1,3].

The recycling of waste PET bottles not only solves the problem of solid waste disposal and recycling, but also its depolymerized products can be used in coating, textile, ink and other industrial fields [1,2,4]. The waste PET materials are treated in the following ways: landfill disposal, mechanical recycling, chemical recycling and incineration [1,5]. Chemical recycling of waste PET is one of the most efficient methods for recycling [2,4]. Chemical recovery can not only decompose polymers into oligomers and monomers, but also oligomers and monomers can be used in new industrial fields, which is not dependent on oil resources and crude oil [1,6].

PET belongs to the polyester family, so the ester groups are easy for chemical degradation [1]. Currently, the ester bonds in waste PET can be hydrolysed [1,7], alcoholysed [1,8], glycolysed [1,9] and ammonolysed [1,10]. Among them, glycolysis is an important and efficient method for chemical recovery of waste PET, such as ethylene glycol (EG) [11], dipropylene glycol (DPG) [12], trimethylpropane (TMP) [1,13], neopentyl glycol (NPG) [1,14] and pentaerythritol (PE) [1,15] had been applied for glycolysis of waste PET.

The glycolysis products of waste PET can be used as raw materials for the coating industry. For example, bis(2-hydroxyethyl) terephthalate (BHET) containing free hydroxyl groups [13,16] can be obtained through the glycolysis of waste PET, and polyols and part of polyacids can be replaced to synthesize polyurethanes [14,15,17], epoxy resin [17–19], saturated or unsaturated polyesters [5,20] and alkyd resins [4,13].

Alkyd resin is a fatty acid modified polyester synthesized by polycondensation reaction of fatty acid, polyol, polybasic acid and monobasic acid, which has attracted more and more attention due to its cheap raw materials, excellent adhesion, high gloss and remarkable wettability [2,4]. Alkyd resin has always occupied a considerable proportion in the field of traditional coatings [2]. There has been much literature on how to use glycolysis products of waste PET to synthesize alkyd resins. For example, alkyd resins based on the glycolysis products of waste PET had been synthesized and corresponding alkyd-amino baking paint had also been explored by Torlakoğlu and Güçlü [21]. Besides, high-performance alkyd resins had been prepared from glycolysis products of different polyols [22]. Moreover, alkyd resins had also been prepared with hyperbranched polyester and glycolysis products of waste PET, and the effects of the fatty acid types, glycolysis products and polyol functionality on thermal properties and film properties had been studied [23].

Up to now, the glycolysis products of waste PET have been widely used to synthesize alkyd resins, but the effect of PET content on the properties of alkyd resins and alkyd films has rarely been explored. Therefore, in this paper, the product of the glycolysed waste PET and low-cost tall oil fatty acids had been used to synthesize autoxidative water-reducible alkyd resins to study the effects of different PET glycolysis products contents on the properties of alkyd resins and films.

Therefore, this research is of great significance for reducing the pollution of PET and developing the preparation technology of water-based alkyd resin based on waste PET as raw material.

# 2. Experimental procedure

## 2.1. Materials

All raw materials used were obtained from the commercial market, except for the waste PET. The waste PET were obtained from discarded soft drink bottles and then they were cut into small pieces, each approximately 5 mm in size [1,4]. The flakes of discarded PET were dipped and washed with distilled water, ethanol and acetone and dried at 100°C for 8 h. PE, phthalic anhydride (PA), trimellitic anhydride (TMA), benzoic acid (BA), EG and zinc acetate were purchased from Shanghai Aladdin Bio-Chem Technology Co., Ltd. Trimethylolpropane (TMP) was supplied by Rous Reagent. Tall oil fatty acid (TOFA) was purchased from Anhui refined oil and fat Co., Ltd. Triethylamine (TEA) and butyl glycol ether (BCS) were obtained from Jiangxi Pinghai Biotechnology Co., Ltd. Titanium dioxide (rutile type) was obtained from Du Pont. The dispersing agent BYK-190, anti-settling agent BYK-420,

**Figure 1.** The pathway of PET glycolysis.

**Table 1.** The feed compositions of the prepared alkyd resins. The content of pentaerythritol (PE), the glycolysis products of waste PET and PET content are highlighted in bold.

| materials (g) | alkyd resins number | | | | | |
| --- | --- | --- | --- | --- | --- | --- |
| | AR 1 | AR 2 | AR 3 | **AR 4** | **AR 5** | **AR 6** |
| TOFA | 80.9 | 80.9 | 80.9 | 80.9 | 80.9 | 80.9 |
| PE | **27.82** | **17.03** | **11.35** | **5.68** | **0** | **0** |
| PA | 58.09 | 58.09 | 58.09 | 58.09 | 58.09 | 58.09 |
| the glycolysis products of waste PET | **0** | **38** | **58** | **68** | **78** | **88** |
| BA | 3.3 | 3.3 | 3.3 | 3.3 | 3.3 | 3.3 |
| EG | 25.4 | 15.52 | 10.36 | 10.36 | 10.36 | 5.18 |
| TMA | 22 | 22 | 22 | 22 | 22 | 22 |
| K | 0.99 | 0.95 | 0.94 | 0.94 | 0.95 | 0.93 |
| R | 1.14 | 1.08 | 1.06 | 1.04 | 1.03 | 1.02 |
| oil length (%) | 41.60 | 41.02 | 40.49 | 40.36 | 40.18 | 39.91 |
| PET content (%) | **0** | **4.1** | **6.0** | **6.9** | **7.7** | **8.5** |

anti-foaming agent BYK-024 and levelling agent BYK-381 were purchased from BYK additives & instruments.

## 2.2. Glycolysis of waste PET

In this experiment, 48 g of waste PET flakes, equivalent to 0.24 mol of the repeating unite in the PET chain, 1.00 mol of trimethylolpropane (TMP) equivalent to 93.8 g, 0.01 mol of zinc acetate equivalent to 1.83 g, used as a catalyst for glycolysis of waste PET flakes, were added into a 500 ml four-necked reactor equipped with a thermometer, reflux condenser, nitrogen inlet and mechanical stirrer. The mixture of glycolysis reaction was heated at 220°C for 6 h in the presence of nitrogen atmosphere protection. As the reaction proceeds, the flakes of discarded PET began to melt, gradually converting from a clear solution to a milky liquid. Then the instruments were removed and the white opaque solid was poured into the can for storage when the glycolysis products of waste PET were cooled to room temperature. The glycolysis pathway of waste PET is shown in figure 1.

## 2.3. Synthesis of water-reducible alkyd resins

The feed compositions employed in this work are listed in table 1, the synthesis reaction equation of water-reducible alkyd resin is shown in figure 2, The short oil water-reducible alkyd resins were prepared by TOFA, PA, PE, EG, BA, TMA and product of the glycolysed PET that include terephthalate of trimethylolpropane, EG and unreacted trimethylolpropane [24,25]. The alkyd constant (K constant) ranged from 0.93 to 0.99 and the ratio of total –OH to total –COOH (R value) was between 1.02 and 1.14. Note, during calculation, trimethylolpropane was used to replace the glycolysed PET according to the feeding ratio.

All raw materials (apart from TMA) were added to the four-neck reactor, which was equipped with a stirrer, thermometer, nitrogen inlet, oil/water separator and rubber stopper. The preparation process of autoxidation alkyd resin is as follows.

*Step 1*. The TOFA, PA, PE, EG, BA, glycolysis products of waste PET and xylene (acts as an azeotropic solvent) were added to the reactor and started to heat slowly until the temperature reaches 180°C.

*Step 2*. When the reaction was kept at 180°C for 1 h, the reaction temperature continued to be slowly raised to 225–230°C, and remained at this temperature until the acid number (AN) of basic alkyd resin was less than

**Figure 2.** The synthesis reactions of water-reducible alkyd resin.

15 mg KOH g$^{-1}$. The AN was titrated by 0.1 M potassium hydroxide solution with phenolphthalein as titration indicator using basic alkyd resin samples dissolved in xylene and ethanol mixed solution.

*Step 3*. When the acid value of the basic alkyd resin was lower than 15 mg KOH g$^{-1}$, the reaction temperature was reduced to 180°C, and TMA was added at this temperature. The reaction was continued until the AN of the alkyd resin was between 40 and 60 mg KOH g$^{-1}$, and then the reaction was completed.

Finally, the alkyd resin was cooled to 80°C, butyl cellosolve was applied to diluted resin to a solid content of 80%, and the corresponding amount of triethylamine neutralizer was also added to the alkyd resin. Therefore, the water-reducible alkyd resin can be prepared by adding water to the above-mentioned alkyd resin.

## 2.4. Preparation of autoxidative water-reducible alkyd coating

The pigment slurry formula is as follows, the mass per cent of rutile titanium dioxide is 75%, deionized water is 20%, dispersant (BYK-190) is 4%, defoamer (BYK-024) is 0.6%, and anti-sedimentation agent (BYK-420) is 0.4%. Meanwhile, the pigment pastes were prepared according to the following process: rutile titanium dioxide and deionized water were ground evenly by the sand mill. And then, BYK-190, BYK-024 and BYK-420 were added to the mixing tank and continuous grinding and dispersion performed until the fineness of titanium dioxide (TiO$_2$) was less than 20 μm.

Table 2 lists the formulations of autoxidative water-borne alkyd coating. The preparation process of the coating consists of two stages. The first step is to disperse the water-reducible alkyd resin with waste PET placed in a stirring tank through a disperser at a speed of 600 r.p.m., and a small amount of BCS was applied as a diluent to reduce the viscosity of the alkyd resin, and then triethylamine was applied to adjust the pH of the resin to 7–8. Next, the drier (OXY-Coat 1101) was added to the mixture and continued to disperse for 10 min.

In the second stage, the pigment pastes and the water-reducible alkyd resin were mixed and evenly dispersed, the viscosity of the system was adjusted to proper viscosity by distilled water, and the solid content of the mixed system was reduced to approximately 50%. Then, the levelling agent BYK-381 and the wetting agent TEGO-245 were added to the mixture system at a low speed (600 r.p.m.) and dispersed for 20 min.

The water-reducible alkyd resins film was prepared by spraying on tinplate, and the reference alkyd resin film was also prepared in the same way. All the prepared films were auto-oxidative cured at 25°C and 50% relative humidity for 7 days.

## 2.5. Characterization

### 2.5.1. Acid number determination of alkyd resin

The AN of alkyd resin was determined according to ASTM D3644. The AN was determined by titration of the sample dissolved in toluene-ethanol solution using 0.1 M potassium hydroxide

**Table 2.** The formulation of autoxidative water-reducible alkyd coating.

| materials | brand | weight (g) |
|---|---|---|
| pigment pastes (75%) | made in laboratory | 30.0 |
| water-reducible alkyd resin (80%) | made in laboratory | 31.3 |
| distilled water | made in laboratory | 38.4 |
| BCS | Jiangxi Pinghai Biotechnology Co., Ltd | 2.0 |
| TEA | Shanghai Aladdin Bio-Chem Technology Co., Ltd. | 2.5 |
| drier | OXY-Coat 1101 | 0.2 |
| wetting agent | TEGO-245 | 0.2 |
| levelling agent | BYK-381 | 0.2 |

solution. From the volume of potassium hydroxide solution obtained by titration, the AN of the sample can be calculated by using

$$AN = \frac{V_{NaOH} \times c_{NaOH} \times 56.1}{m_{smaple}}.$$

### 2.5.2. Viscosity determination of water-reducible alkyd resin

The viscosity of water-reducible alkyd resin was determined through the BEVS1132 Cone viscometer from BEVS, Inc., according to the standard test method ASTM D1545-89.

### 2.5.3. Particle size analysis

To determine the particle size of the water-reducible alkyd resins, the resins were diluted with distilled water to approximately 1% and Nicomp 380DLS of the Malvern Instrument was used at room temperature.

### 2.5.4. GPC analysis

The molecular weight of water-reducible alkyd resins base on waste PET was determined by Waters 1515 gel permeation chromatograph (GPC). All samples dissolved in tetrahydrofuran (THF) of HPLC grade, the insoluble components were filtered and the filtrate retained as the mobile phase. The flow rate of the mobile phase was set at 1.0 ml min$^{-1}$, and the molecular weight of alkyd resin was calculated by using polystyrene as the reference standard at 25°C.

### 2.5.5. Testing of water-reducible alkyd resin films

In order to estimate the properties of the water-reducible alkyd coatings, the following standards will be used to test the coatings.

According to the standard test method ASTM D3363, the hardness of the water-reducible alkyd coatings was measured by the BGD506/1 pencil hardness tester.

The adhesion of water-reducible alkyd coatings was measured by using the BEVS2202 cross hatch cutter from BEVS, Inc., according to the standard test method ASTM D3359.

The gloss of water-reducible alkyd coatings was measured at an angle of 60° (ASTM D2457) using a gloss meter (BEVS 60° Glossmeter from BEVS industrial Co., Ltd).

The flexibility of water-reducible alkyd coatings was measured according to ASTM D522 using a cylindrical mandrel bend tester (BEVS1603 cylindrical mandrel bend tester from BEVS industrial Co., Ltd).

The impact test of water-reducible alkyd coatings was measured by using the BEVS impact tester from BEVS, Inc., according to the standard test method ASTM D2794.

The alkaline, acid and water resistance of water-reducible alkyd coatings were tested by using a lump of cotton to absorb 5% sodium carbonate solution, 2% sulfuric acid solution and distilled water by immersing on the coatings, according to the standard test methods ASTM D1647 and ASTM D1647-59, respectively.

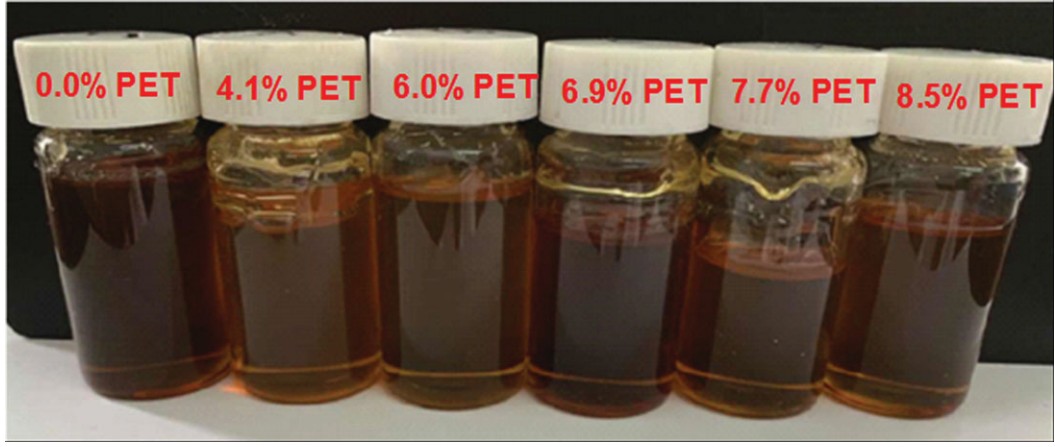

**Figure 3.** The appearance of water-reducible alkyd resins with different content of the glycolysed PET in BCS (80% solid content).

# 3. Results and discussion

As we know, the oil content in the resin will affect the hardness, drying rate, gloss and other performance properties of coatings [2,25]. Short oil alkyd coatings have poor water and chemical resistance, but the drying rate and hardness of the coatings are better than medium or long oil alkyd resin films [8,23].

However, the shorter the oil length of the alkyd resins, the easier it is to gel during the preparation process [2,23]. In order to prevent premature gelation in the synthesis of alkyd resins, it is necessary to use an excessive number of polyols [4,9]. Therefore, in this experiment, the alkyd constant (K) and ratio of total –OH groups to total –COOH groups (R) were set to 0.93–0.99 and 1.02–1.14, respectively. In addition, owing to part of the polyols and polyacids of the alkyd resins being replaced by the glycolysed products of waste PET, the oil length in this experiment was between 41.60% and 39.91%.

## 3.1. Characterizations of the alkyd resin

The appearance of alkyd resins (dissolved in BCS, solid content is approximately 80%) with different content of glycolysed PET is shown in figure 3. It can be seen from figure 3 that the alkyd resin synthesized from glycolysed PET has a brownish-yellow appearance and was a viscous fluid substance at room temperature. This was due to the TOFA used in the synthesis process, which had carbon–carbon double bonds in its molecular structure. In the presence of high temperature and oxygen, it is easy to cause the double bond to be oxidized and the resin colour becomes darker [9,26].

Particle size is an important index of water-borne resin stability or dispersion. The average particle size of water-reducible alkyd resin not only affects the dispersion effect of alkyd molecules in the medium, but also influences the storage stability of alkyd coating [9,21]. As illustrated from figure 4*a*, there are no obvious differences in transparency and light blue opalescence in water-reducible alkyd resins (1% solid content) with different content of glycolysed products of waste PET. Additionally, it can be seen from figure 4*b* that the particle size of water-reducible alkyd resins based on glycolysed waste PET was approximately 100 nm, indicating that the resin can be dispersed in aqueous solution in the form of small soft colloidal particles [9,27]. It can also be seen from figure 4*b* that the particle size of the water-reducible alkyd resin without glycolysis product of waste PET was larger than those of the resins with glycolysis product of waste PET. The author speculates that the reasons for this result may be that with the increase of PET content, under the condition of a certain amount of TMA was added, the shorter the oil length, the greater the hydrophilicity of the water-reducible alkyd resin, another reason is that the water-reducible alkyd resin without PET contains more PE with greater steric hindrance. However, when the oil length continues to decrease and the PET content continues to increase, the particle size does not decrease linearly. This is the result of a combination of factors that include the steric hindrance effects of the PET glycolysed product and PE.

Viscosity was a measure of preventing liquid flow. Since alkyd resins usually have longer molecular chains, the molecular chains of alkyd molecules tend to entangle with each other at a certain temperature. This entanglement structure was different from the chemical cross-linking between polymer molecular.

(*a*)

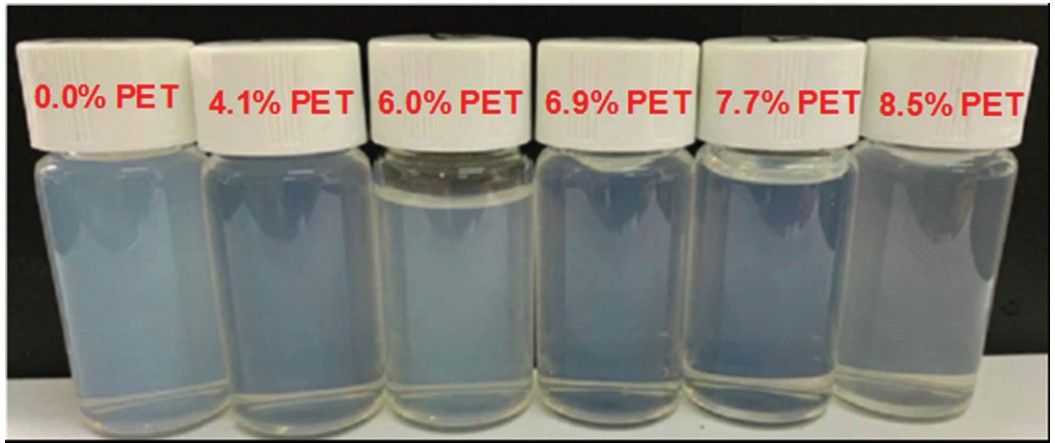

(*b*)

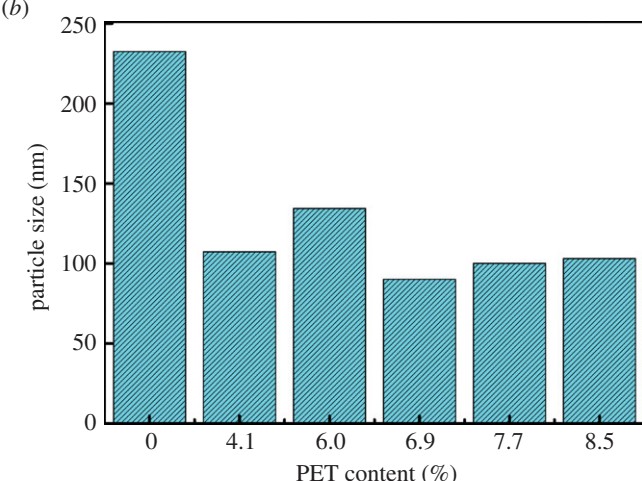

**Figure 4.** The appearance (*a*) and particle size (*b*) of water-reducible alkyd resin with different content of the glycolysed PET in water (1% solid content).

This kind of molecular chain entanglement was formed by the intermolecular force or the mutual entanglement of molecular chains. However, viscosity is not only affected by intermolecular forces, but also related to the raw materials used in the resin synthesis processing [4,17,28].

As can be seen from figure 5, the viscosity of alkyd resins containing the glycolysed PET is significantly higher than that without PET. This can be attributed to three reasons. On the one hand, the shorter the oil length, the greater the viscosity [22,29]. On the other hand, it is because the $T_g$ of the alkyd resin based on the glycolysed PET (contains terephthalic acid) is higher than those of the alkyd synthesized from PA [30]. Finally, in water-reducible resins, the smaller the particle size, the stronger the intermolecular forces, resulting in greater viscosity.

The molecular weight of alkyd resin is an important factor that determines the performance of the films. The molecular weight of alkyd resins with different content of the glycolysed PET was determined by gel permeation chromatography, and the results are shown in table 3. The results show that the $M_n$ of the water-reducible alkyd resins was between 2300 and 2900, with little fluctuation. The weight average molecular weight was a component of the polymer chain structure and an important indicator of the size of the macromolecule [14,17]. The mechanical properties and processing properties of the film will be affected by the weight average molecular weight. If the weight average molecular weight was lower than the statistical average value, the alkyd resin film will have no mechanical strength [30]. For example, as the weight average molecular weight increases, the mechanical properties of the film will increase. The polydispersity index of a polymer can simply describe the distribution width of molecular weight, and the performance of a resin film through the distribution width can be roughly judged. Generally, the smaller the polydispersity index, the narrower the molecular weight distribution, which makes the film excellent in terms of mechanical properties [31]. The alkyd resin synthesized in this

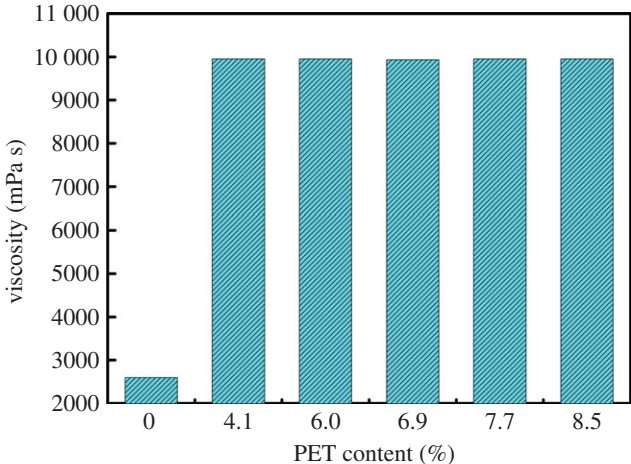

**Figure 5.** Effect of the content of the glycolysed PET on the viscosity of water-reducible alkyd resin (80% solid content).

**Table 3.** The molecular weight of alkyd resins with different content of glycolysed waste PET products.

| entry | alkyd resin number | | | | | |
|---|---|---|---|---|---|---|
| | AR 1 | AR 2 | AR 3 | AR 4 | AR 5 | AR 6 |
| $M_n$ | 2757 | 2835 | 2899 | 2694 | 2662 | 2341 |
| $M_w$ | 7909 | 10 675 | 11 235 | 7174 | 7990 | 5357 |
| PDI | 2.86 | 3.76 | 3.87 | 2.66 | 3.00 | 2.28 |

**Table 4.** Physical properties of the water-reducible alkyd resin coatings.

| testing | alkyd resin number | | | | | | |
|---|---|---|---|---|---|---|---|
| | AR 1 | AR 2 | AR 3 | AR 4 | AR 5 | AR 6 | AR-Re |
| thickness (µm) | 20.8 | 20.3 | 20.0 | 21.1 | 26.6 | 22.0 | 21.3 |
| pencil hardness | HB | H | H | H | H | H | HB |
| adhesion | 5 | 5 | 5 | 5 | 5 | 5 | 5 |
| gloss (60°) | 90.3 | 87.0 | 95.5 | 89.5 | 90.1 | 89.5 | 89.3 |
| flexibility (mm) | 1 | 1 | 1 | 1 | 1 | 1 | 2 |
| impact test (cm) | 50.0 | 50.0 | 50.0 | 50.0 | 50.0 | 50.0 | 50.0 |

experiment has a small molecular weight distribution and a stable number average molecular weight. Therefore, an autoxidative cross-linking mechanism of the prepared water-reducible alkyd with the glycolysed PET would bringing better film properties [31,32].

## 3.2. Physical properties of the water-reducible alkyd resin films

The coatings of water-reducible alkyd resins were prepared from different content of the glycolysed PET by spraying on tinplate. In this paper, commercial alkyd resin was used as a reference resin (AR-Re) to compared with the alkyd resins synthesized in this experiment. The physical properties of the water-reducible alkyd resin coatings are given in table 4.

In order to achieve long-term protection of the substrate, the films need excellent adhesion. On the metal substrate, the adhesion of the films can isolate oxygen and inhibited the formation of rust

[22,28]. Even if the films encounter physical damage in daily life, the films will not leave the surface of the substrate. The adhesion of the water-reducible alkyd films to the tinplate substrate was tested by the cross-hatch adhesion method [33]. The results of adhesion test (table 4) demonstrate that the water-reducible alkyd films have excellent adhesion and can firmly adhere to the surface of the metal substrate. This excellent performance may be originated in chemical and physical reasons. On the one hand, because the tinplate needs to be pre-treated and polished, the surface of the substrate was porous. Additionally, the excellent wetting properties of the alkyd films, the grooves caused by the polishing will be filled, which was essentially a mechanical anchor between the metal substrate and the films. On the other hand, the alkyd resin itself can be firmly attached to the surface of the substrate through cross-linking [34–36].

The films will experience many different deformations throughout their service life. These deformations must be withstood without performance damage, physical damage or loss of adhesion. Therefore, this requires the film to possess outstanding flexibility and impact resistance to face mechanical deformations in life [36]. In the process of testing the flexibility of the film with the cone bending instrument, the film of the reference resin (AR-Re) began to collapse at 2 mm, and all the alkyd resins synthesized in our experiment can pass the bending flexibility test of 1 mm without any film problem. This shows that the alkyd resin film was relatively flexible, and can have resistance to one-dimensional deformation without film damage. In addition, when determining the elastic properties of the films, it is necessary to distinguish between bending elasticity (slow deformation) and impact elasticity (fast deformation) [3]. The alkyd resin films prepared in this experiment can withstand an impact of 50 cm, and the films will not crack or peel after the impact. This shows that the films also possess excellent resistance to three-dimensional deformation or fast deformation. Therefore, the alkyd resins synthesized by glycolysis product of waste PET in this experiment have excellent flexibility and impact resistance.

In addition to excellent adhesion, flexibility and impact resistance, the hardness of paint film is also important. For all alkyd resins prepared from glycolysed products, the pencil hardness of the films can reach H grade. This may be because the water-reducible alkyd films prepared from glycolysed waste PET contain a large number of oligomers of bis(2-hydroxyethyl) terephthalate, and these oligomers possess rigid aromatic ring structure, which increases the rigidity and produces high hardness. Therefore, the introduction of aromatic rings into the backbone of alkyd resin will increase the rigidity of the molecular chain [4,21,30].

The gloss of alkyd resin films is an evaluation index for its decoration and aesthetics. If the particle size of the resin is smaller, the surface of the film will be smoother than the surface of the resin with a larger particle size after cross-linking to form a film, which can reflect incident light more effectively, reduce scattering and improve the gloss of the film [12,30,36]. As shown in table 4, the 60° gloss of all alkyd resin films was approximately 85–90, which was related to the surface of the resin being smooth after cross-linking.

At the same time, the alkyd resins synthesized in this experiment were compared with commercial alkyd resin. It can be seen from the results in table 4 that the film performances of the alkyd resin synthesized from the glycolysis product of waste PET were equivalent to those of the commercial alkyd resin, but the film hardness of the former was higher than that of the latter.

## 3.3. Chemical resistance properties of the water-reducible alkyd resin films

Chemical resistance means that the film can withstand a specific concentration of chemical solutions and the damage to the chemical reactions. Since the film mainly protects the covered objects for a long time, it was often exposed to wind, sun, rain, water immersion and various types of corrosion. Therefore, the durability of alkyd resins was tested through chemical resistance [1,22,24].

In normal use, alkyd resins are rarely affected by acidic solutions. It can be found in table 5 that the acid resistance of alkyd films prepared from glycolysis product of waste PET was reduced. It presumed that this was because the alkyd resin prepared with the waste PET product of glycolysis had a smaller particle size and was denser when forming a film [33,36]. The hydrogen generated after the sulfuric acid solution penetrates the film and contacts the tinplate cannot easily penetrate the film and cause blistering.

For alkyd resins, water resistance and alkali resistance play a crucial role in order to achieve long service life and durability. Table 5 demonstrates that the alkyd resin films synthesized from the glycolysis product of waste PET have excellent water resistance. After 24 h of soaking in water, with the increase in the amount of glycolysis product of waste PET, the water resistance of the film gradually increases. Due to the different chemical structures of the raw materials, the difference in

**Table 5.** Chemical resistance properties of the water-reducible alkyd resin films. NC: no change. Effect of soaking wet cotton with 5% Na₂CO₃ solution on film. Effect of soaking wet cotton with 2% H₂SO₄ solution on film. Effect of soaking wet cotton with water on film.

| alkyd resins number | testing | | |
|---|---|---|---|
| | alkaline resistance (24 h) | acid resistance (24 h) | water resistance (24 h) |
| AR 1 | slight blister | NC | slight blister |
| AR 2 | whiten, blister | slight blister | whiten, slight blister |
| AR 3 | whiten, blister | slight blister | whiten, slight blister |
| AR 4 | whiten, blister | blister | whiten |
| AR 5 | whiten, blister | severe blister | slight whiten |
| AR 6 | whiten, blister | severe blister | NC |
| AR-Re | severe blister | whiten, blister | NC |

water resistance will be manifested. Since there are a large number of terephthalic acid ester structures in glycolysis products, compared with PA, the ester bond at the para-position shows better hydrolysis resistance [30].

According to table 5, the poor alkaline resistance of water-reducible alkyd resin film may be due to the fact that a large number of ester bonds in its molecular backbone were hydrolysed under alkaline conditions.

## 4. Conclusion

The waste PET was glycolysed by trimethylolpropane with zinc acetate as catalyst. Low-cost and environmentally friendly bio-based water-reducible alkyd resins with different content glycolysis product of waste PET were prepared, and the corresponding coating properties were investigated. Meanwhile, experimental results had been compared with commercial water-reducible alkyd and similar water-reducible alkyd without waste PET. The following conclusions can be drawn from the obtained results.

The maximum concentration of PET in autoxidative water-reducible alkyd resins can reach 8.5 wt%, and the molecular weight, particle size and viscosity of water-reducible alkyd resins do not change much with the increase of PET concentration. The introduction of PET resulted in the viscosity of water-reducible alkyd resins being greater than that of water-reducible alkyd resin without PET; this is mainly because PET contains harder terephthalic acid monomer units and smaller particle size for water-reducible alkyd with PET. (In water-reducible resins, the smaller the particle size, the stronger the intermolecular forces, resulting in greater viscosity.) Indeed, the particle size of water-reducible alkyd resins with waste PET is significantly lower than that of the water-reducible alkyd resin without PET, this is due to PET-free water-reducible alkyd resin contain more PE with greater steric hindrance.

In addition, the physical properties (gloss, adhesion, hardness, flexibility and impact resistance) and chemical resistance properties of autoxidative water-reducible alkyd coatings in this work is comparable to those of alkyd coating prepared with commercial water-reducible alkyd and similar water-reducible alkyd without waste PET.

Therefore, the recycling of waste PET can not only reduce the production cost of alkyd resin, but also eliminate the pollution of waste PET.

Data accessibility. Our data are deposited at Dryad Digital Repository: https://doi.org/10.5061/dryad.nvx0k6dr0.
Authors' contributions. Y.D. and L.S. designed the study. S.O., Y.X. and W.F. prepared all samples for analysis. S.O. and Y.D. collected and analysed the data. S.O., Y.D. and L.S. interpreted the results and wrote the manuscript. All authors gave final approval for publications.
Competing interests. We declare we have no competing interests.
Funding. We received no funding for this study.

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
