## [Peer Review File · Royal Society Open Science]

Review History

RSOS-202375.R0 (Original submission)

Review form: Reviewer 1

Is the manuscript scientifically sound in its present form?

No

Are the interpretations and conclusions justified by the results?

No

Is the language acceptable?

Yes

Do you have any ethical concerns with this paper?

No

Have you any concerns about statistical analyses in this paper?

No

Recommendation?

Reject

Comments to the Author(s)

Please find here the comment for the manuscript RSOS-202375 entitled "Preparation of autoxidative water-borne alkyd resin films from waste polyethylene terephthalate". In this manuscript, authors had been studied and reported the effects of different content glycolysis products of waste PET on the appearance, viscosity, particle size, and molecular weight of autoxidative water-borne alkyd resins and the corresponding film adhesion, flexibility, impact resistance, gloss, hardness, and chemical resistance. The Manuscript topic is good but the author unable to prove its worth, there is no experimental evidence of whether the desired product is formed or not and the should be some more experimental studies to prove the worth of the manuscript. The manuscript is not properly written and has some serious language, presentation, and novelty issues that may be justified before its publication So I recommend rejection of this manuscript for its publication in "Royal Society Open Science".

Regard

Review form: Reviewer 2

Is the manuscript scientifically sound in its present form?

Yes

Are the interpretations and conclusions justified by the results?

Yes

Is the language acceptable?

Yes

Do you have any ethical concerns with this paper?

No

Have you any concerns about statistical analyses in this paper?

No

Recommendation?

Accept with minor revision (please list in comments)

Comments to the Author(s)

In this paper, the authors prepared a series of autoxidative water-borne alkyd resins with different contents of the glycolysis product from soft PET bottles and determined the properties of the as-obtained alkyd resins and the performances of the cured coatings. The work could further contribute to deeply understand the effect of the glycolysis product of PET on its based alkyd coatings, and hence facilitating the application the recycling product of PET in coatings industry. I think the paper is publishable in Royal Society Open Science.

Prior to publication, I suggest the authors carry out the following minor revisions:

- (1) The abstract only described the change trends of the properties of the alkyd resin and its based coatings after incorporation of the glycolysis product of PET. The main reasons for these trends had better also be explained, which can increase the academic level of the manuscript.
- (2) The conclusion section is a bit long. Please shorten it.
- (3) In Scheme 1, the ethylene glycol unit in PET disappeared after glycolysis. What does this unit transform into after glycolysis?
- (4) Some minor grammar or editing mistakes:
 - (a) Please define the full name of PET in the abstract section.

- (b) Page 3, line 1-2 in the right column, “acetate was used as a catalyst for glycolysis of waste PET flakes, were carried out in a 500 mL four-necked “ should be ““ acetate equivalent to 1.83 g, used as a catalyst for glycolysis of waste PET flakes, were added into a 500 mL four-necked”.
- (c) Please clearly define the composition of “the glycolysis products of waste PET” in Table 2 or describe where it come from.
- (d) Page 5, line 49 in the left column, “styrene” should be “polystyrene”.
- (e) In Table 3, the significant digit of PDI is excessive. One or two decimal is enough.
- (f) Page 7, line 56 in the left column, “the weight average molecular” should be “the weight average molecular weight”.
- (g) Page 8, line 23 in the left column, “comparede” should be “compared”.

Review form: Reviewer 3

Is the manuscript scientifically sound in its present form?

Yes

Are the interpretations and conclusions justified by the results?

Yes

Is the language acceptable?

Yes

Do you have any ethical concerns with this paper?

No

Have you any concerns about statistical analyses in this paper?

No

Recommendation?

Accept with minor revision (please list in comments)

Comments to the Author(s)

1. Authors should add more data regarding glycolysis and add more results (numerical values) in the abstract.
2. Page 2 line 35. Landfill disposal and incineration are not types of recycling but of waste management. Please rephrase this sentence.
3. Page 3. Did authors analyzed product of glycolysis (acid and hydroxyl number) or just used theoretical values for the calculation of feed composition for alkyd syntheses? In some cases some amount of EG could be lost due to evaporation which lowers OH value of glycolysed product. This could further affect resin properties.
4. Page 4 line 5 right column. Change triphthalic anhydride to trimellitic anhydride.
5. In Scheme 2 authors presented TMA as if COOH group reacted. Isn't it more likely that anhydride group reacted?
6. Page 7. Authors explained that oil length affected particle size. The oil length decreases linearly with the increase in glycolysed PET amount, while particle size is higher only for the alkyd without waste PET while for the other samples is similar. If the oil length is the only factor than particle size would decrease with the increase in PET content. Authors should consider other factors too.
7. Page 7. In my opinion the main reason why viscosity of resin without PET is significantly lower is due to the larger particle size. I agree with the Authors that the oil length affects resin viscosity but considering just oil length it is hard to explain why neat resin shows significantly lower

viscosity. In my opinion authors should find a reason why neat resin had higher particle size and attribute this to lower viscosity. Maybe, if authors did not analyzed product of PET glycolysis, and only used theoretical values of -OH and -COOH numbers, this could be a reason. For example if some of EG was lost this could derange calculations for resin feed composition 8. Page 7 line 56. Authors wrote " It can be expected that when the autoxidation water-borne alkyd resin film was cross-linked, it would have a larger number average molecular weight and mass average molecular weight, as well as better film properties." This is not something that "could be expected" but it is. After crosslinking Mn and Mw increases so high that it is no longer viable to talk about molecular weight.

9. Conclusion sounds like short list of Results. Authors should add short explanations why did they get these results and add why resins with PET showed smaller particle size which affected viscosity.

Decision letter (RSOS-202375.R0)

Dear Dr Yong-bo:

Title: Preparation of autoxidative water-borne alkyd resin films from waste polyethylene terephthalate
Manuscript ID: RSOS-202375

The editor assigned to your manuscript has now received comments from reviewers. We would like you to revise your paper in accordance with the referee and Subject Editor suggestions which can be found below (not including confidential reports to the Editor). Please note this decision does not guarantee eventual acceptance.

Please submit your revised paper before 04-Apr-2021. Please note that the revision deadline will expire at 00.00am on this date. If we do not hear from you within this time then it will be assumed that the paper has been withdrawn. In exceptional circumstances, extensions may be possible if agreed with the Editorial Office in advance. We do not allow multiple rounds of revision so we urge you to make every effort to fully address all of the comments at this stage. If deemed necessary by the Editors, your manuscript will be sent back to one or more of the original reviewers for assessment. If the original reviewers are not available we may invite new reviewers.

When submitting your revised manuscript, you must respond to the comments made by the referees and upload a file "Response to Referees" in "Section 6 - File Upload". Please use this to document how you have responded to the comments, and the adjustments you have made. In

order to expedite the processing of the revised manuscript, please be as specific as possible in your response.

On behalf of the Subject Editor Professor Anthony Stace and the Associate Editor Professor Chaohua Cui.

RSC Associate Editor:
Comments to the Author:
(There are no comments.)

RSC Subject Editor:
Comments to the Author:
(There are no comments.)

Reviewers' Comments to Author:

Reviewer: 1

Comments to the Author(s)

Please find here the comment for the manuscript RSOS-202375 entitled "Preparation of autoxidative water-borne alkyd resin films from waste polyethylene terephthalate". In this manuscript, authors had been studied and reported the effects of different content glycolysis products of waste PET on the appearance, viscosity, particle size, and molecular weight of autoxidative water-borne alkyd resins and the corresponding film adhesion, flexibility, impact resistance, gloss, hardness, and chemical resistance. The Manuscript topic is good but the author unable to prove its worth, there is no experimental evidence of whether the desired product is formed or not and the should be some more experimental studies to prove the worth of the manuscript. The manuscript is not properly written and has some serious language, presentation, and novelty issues that may be justified before its publication So I recommend rejection of this manuscript for its publication in "Royal Society Open Science".

Regard

Reviewer: 2

Comments to the Author(s)

In this paper, the authors prepared a series of autoxidative water-borne alkyd resins with different contents of the glycolysis product from soft PET bottles and determined the properties of the as-obtained alkyd resins and the performances of the cured coatings. The work could

further contribute to deeply understand the effect of the glycolysis product of PET on its based alkyd coatings, and hence facilitating the application the recycling product of PET in coatings industry. I think the paper is publishable in Royal Society Open Science.

Prior to publication, I suggest the authors carry out the following minor revisions:

- (1) The abstract only described the change trends of the properties of the alkyd resin and its based coatings after incorporation of the glycolysis product of PET. The main reasons for these trends had better also be explained, which can increase the academic level of the manuscript.
- (2) The conclusion section is a bit long. Please shorten it.
- (3) In Scheme 1, the ethylene glycol unit in PET disappeared after glycolysis. What does this unit transform into after glycolysis?
- (4) Some minor grammar or editing mistakes:
 - (a) Please define the full name of PET in the abstract section.
 - (b) Page 3, line 1-2 in the right column, "acetate was used as a catalyst for glycolysis of waste PET flakes, were carried out in a 500 mL four-necked " should be ""acetate equivalent to 1.83 g, used as a catalyst for glycolysis of waste PET flakes, were added into a 500 mL four-necked".
 - (c) Please clearly define the composition of "the glycolysis products of waste PET" in Table 2 or describe where it come from.
 - (d) Page 5, line 49 in the left column, "styrene" should be "polystyrene".
 - (e) In Table 3, the significant digit of PDI is excessive. One or two decimal is enough.
 - (f) Page 7, line 56 in the left column, "the weight average molecular" should be "the weight average molecular weight".
 - (g) Page 8, line 23 in the left column, "comparede" should be "compared".

Reviewer: 3

Comments to the Author(s)

1. Authors should add more data regarding glycolysis and add more results (numerical values) in the abstract.
2. Page 2 line 35. Landfill disposal and incineration are not types of recycling but of waste management. Please rephrase this sentence.
3. Page 3. Did authors analyzed product of glycolysis (acid and hydroxyl number) or just used theoretical values for the calculation of feed composition for alkyd syntheses? In some cases some amount of EG could be lost due to evaporation which lowers OH value of glycolysed product. This could further affect resin properties.
4. Page 4 line 5 right column. Change triphthalic anhydride to trimellitic anhydride.
5. In Scheme 2 authors presented TMA as if COOH group reacted. Isn't it more likely that anhydride group reacted?
6. Page 7. Authors explained that oil length affected particle size. The oil length decreases linearly with the increase in glycolysed PET amount, while particle size is higher only for the alkyd without waste PET while for the other samples is similar. If the oil length is the only factor than particle size would decrease with the increase in PET content. Authors should consider other factors too.
7. Page 7. In my opinion the main reason why viscosity of resin without PET is significantly lower is due to the larger particle size. I agree with the Authors that the oil length affects resin viscosity but considering just oil length it is hard to explain why neat resin shows significantly lower viscosity. In my opinion authors should find a reason why neat resin had higher particle size and attribute this to lower viscosity. Maybe, if authors did not analyzed product of PET glycolysis, and only used theoretical values of -OH and -COOH numbers, this could be a reason. For example if some of EG was lost this could derange calculations for resin feed composition
8. Page 7 line 56. Authors wrote " It can be expected that when the autoxidation water-borne alkyd resin film was cross-linked, it would have a larger number average molecular weight and mass average molecular weight, as well as better film properties." This is not something that

"could be expected" but it is. After crosslinking Mn and Mw increases so high that it is no longer viable to talk about molecular weight.

9. Conclusion sounds like short list of Results. Authors should add short explanations why did they get these results and add why resins with PET showed smaller particle size which affected viscosity.

Author's Response to Decision Letter for (RSOS-202375.R0)

See Appendix A.

RSOS-202375.R1 (Revision)

Review form: Reviewer 2

Is the manuscript scientifically sound in its present form?

Yes

Are the interpretations and conclusions justified by the results?

Yes

Is the language acceptable?

Yes

Do you have any ethical concerns with this paper?

No

Have you any concerns about statistical analyses in this paper?

No

Recommendation?

Accept as is

Comments to the Author(s)

The authors have responded to reviewer's comments adequately. I think it is publishable now.

Review form: Reviewer 3

Is the manuscript scientifically sound in its present form?

Yes

Are the interpretations and conclusions justified by the results?

Yes

Is the language acceptable?

Yes

Do you have any ethical concerns with this paper?

No

Have you any concerns about statistical analyses in this paper?

No

Recommendation?

Accept as is

Comments to the Author(s)

Paper is now suitable for publication.

Decision letter (RSOS-202375.R1)

Dear Dr Yong-bo:

Title: Preparation of autoxidative water-reducible alkyd resins from waste polyethylene terephthalate

Manuscript ID: RSOS-202375.R1

It is a pleasure to accept your manuscript in its current form for publication in Royal Society Open Science. The chemistry content of Royal Society Open Science is published in collaboration with the Royal Society of Chemistry.

On behalf of the Subject Editor Professor Anthony Stace and the Associate Editor Professor Chaohua Cui.

RSC Associate Editor:
Comments to the Author:
(There are no comments.)

RSC Associate Editor:
Comments to the Author:
(There are no comments.)

Reviewer(s)' Comments to Author:
Reviewer: 2

Comments to the Author(s)
The authors have responded to reviewer's comments adequately. I think it is publishable now.

Reviewer: 3

Comments to the Author(s)
Paper is now suitable for publication.

Appendix A

Dear Editors

Thank you for your letter and for the reviewers' comments concerning our manuscript (Manuscript ID: RSOS-202375) entitled "Preparation of autoxidative water-reducible alkyd resins from waste polyethylene terephthalate". Those comments are all valuable and very helpful for revising and improving our paper, as well as the important guiding significance to our researches. We have studied comments carefully and have made correction which we hope meet with approval. Revised portion are marked in red in the revised manuscript.

We greatly appreciate your continued interest in our manuscript.

Best wishes,

Sincerely yours,

Yong-Bo Ding

The main corrections in the paper and the responds to the reviewer's comments are as following:

Responds to the reviewer's comments:

Reviewer #1:

Comments to the Author(s)

Please find here the comment for the manuscript RSOS-202375 entitled "Preparation of autoxidative water-borne alkyd resin films from waste polyethylene terephthalate". In this manuscript, authors had been studied and reported the effects of different content glycolysis products of waste PET on the appearance, viscosity, particle size, and molecular weight of autoxidative water-borne alkyd resins and the corresponding film adhesion, flexibility, impact resistance, gloss, hardness, and chemical resistance. The Manuscript topic is good but the author unable to prove its worth, there is no experimental evidence of whether the desired product is formed or not and the should

be some more experimental studies to prove the worth of the manuscript. The manuscript is not properly written and has some serious language, presentation, and novelty issues that may be justified before its publication So I recommend rejection of this manuscript for its publication in “Royal Society Open Science”.

Response: We are very sorry for our incorrect grammar and serious language problems. These problems have been corrected in the revised manuscript. In the meantime, this work is valuable. For example, the film performances of water-reducible alkyd resins prepared from waste PET are comparable to those of commercial water-reducible alkyd resin and free-PET water-reducible alkyd resin. Therefore, the use of waste PET in water-borne coatings systems not only reduces the cost of coatings, but also opens up a new market for recycled PET, which may contribute a promising method for management of waste PET. Please see the abstract section in the revised manuscript.

Reviewer: 2

Comments to the Author(s)

In this paper, the authors prepared a series of autoxidative water-borne alkyd resins with different contents of the glycolysis product from soft PET bottles and determined the properties of the as-obtained alkyd resins and the performances of the cured coatings. The work could further contribute to deeply understand the effect of the glycolysis product of PET on its based alkyd coatings, and hence facilitating the application the recycling product of PET in coatings industry. I think the paper is publishable in Royal Society Open Science. Prior to publication, I suggest the authors carry out the following minor revisions:

(1) The abstract only described the change trends of the properties of the alkyd resin and its based coatings after incorporation of the glycolysis product of PET. The main reasons for these trends had better also be explained, which can increase the academic level of the manuscript.

Response: Thank you very much for your suggestion concerning the abstract. The main reasons for these trends had been explained, please see the abstract section in

the revised manuscript.

(2) The conclusion section is a bit long. Please shorten it.

Response: Thank you very much for your suggestions for the conclusion section. The problem had been corrected in the revised manuscript. Please see the conclusion section in the revised manuscript.

(3) In Scheme 1, the ethylene glycol unit in PET disappeared after glycolysis. What does this unit transform into after glycolysis?

Response: We are very sorry for our mistake that the ethylene glycol unit in PET disappeared after glycolysis, Scheme 1 had been modified, please see Figure 1 in the revised manuscript.

(4) Some minor grammar or editing mistakes:

(a) Please define the full name of PET in the abstract section.

Response: Thank you for underlining this deficiency. The full name of PET in the abstract section had been defined, please see abstract section in the revised manuscript.

(b) Page 3, line 1-2 in the right column, “acetate was used as a catalyst for glycolysis of waste PET flakes, were carried out in a 500 mL four-necked “ should be ““acetate equivalent to 1.83 g, used as a catalyst for glycolysis of waste PET flakes, were added into a 500 mL four-necked”.

Response: We are very sorry for our incorrect grammar and editing problem. These problems have been corrected in the revised manuscript.

(c) Please clearly define the composition of “the glycolysis products of waste PET” in Table 2 or describe where it come from.

Response: The glycolysis products of waste PET include terephthalate of trimethylolpropane, ethylene glycol and unreacted trimethylolpropane, please see

“Synthesis of water-reducible alkyd resins” section in the revised manuscript.

(d) Page 5, line 49 in the left column, “styrene” should be “polystyrene”.

Response: we are very sorry for our incorrect spelling. According to your suggestion, this error has been corrected in the revised manuscript.

(e) In Table 3, the significant digit of PDI is excessive. One or two decimal is enough.

Response: Thank you for the suggestion, we have revised significant digit of PDI in the revised manuscript.

(f) Page 7, line 56 in the left column, “the weight average molecular” should be “the weight average molecular weight”.

Response: Thank you for your suggestion, the problem have been corrected in the revised manuscript.

(g) Page 8, line 23 in the left column, “comparede” should be “compared”.

Response: We are very sorry for our incorrect spelling; the problem had been corrected in the revised manuscript.

Reviewer: 3

Comments to the Author(s)

1. Authors should add more data regarding glycolysis and add more results (numerical values) in the abstract.

Response: Thank you for your suggestion, the problem have been corrected in the revised manuscript.

2. Page 2 line 35. Landfill disposal and incineration are not types of recycling but of waste management. Please rephrase this sentence.

Response: Thank you for your suggestion, the problem have been corrected in the

revised manuscript.

3. Page 3. Did authors analyzed product of glycolysis (acid and hydroxyl number) or just used theoretical values for the calculation of feed composition for alkyd syntheses? In some cases some amount of EG could be lost due to evaporation which lowers OH value of glycolysed product. This could further affect resin properties.

Response: Yes, in some cases some amount of EG could be lost due to evaporation which lowers OH value of glycolysed product, but in this work, theoretical values were used to the calculation of feed composition for alkyd syntheses due to the theoretical hydroxyl value and the experimental hydroxyl value have little change according to reference (P.M. Spasojević, V.V. Panić, J.V. Džunuzović, A.D. Marinković, A.J.J. Woortman, K. Loos, I.G. Popović, High performance alkyd resins synthesized from postconsumer PET bottles, RSC Advances 5(76) (2015) 62273-62283.). The problem had been explained in the revised manuscript, please see “Synthesis of water-reducible alkyd resins” section in the revised manuscript.

4. Page 4 line 5 right column. Change triphthalic anhydride to trimellitic anhydride.

Response: We are very sorry for our incorrect spelling. The problem had been corrected in the revised manuscript.

5. In Scheme 2 authors presented TMA as if COOH group reacted. Isn't it more likely that anhydride group reacted?

Response: Thank you for underlining this deficiency. We apologize for the mistakes made, this section was revised and modified, please see Figure 2 in the revised manuscript.

6. Page 7. Authors explained that oil length affected particle size. The oil length decreases linearly with the increase in glycolysed PET amount, while particle size is higher only for the alkyd without waste PET while for the other samples is similar. If the oil length is the only factor than particle size would decrease with the increase in

PET content. Authors should consider other factors too.

Response: Yes, the oil length is not the only factor affected particle size, the steric effect is another factor that affects the particle size, please see "Particle size" section in the revised manuscript.

7. Page 7. In my opinion the main reason why viscosity of resin without PET is significantly lower is due to the larger particle size. I agree with the Authors that the oil length affects resin viscosity but considering just oil length it is hard to explain why neat resin shows significantly lower viscosity. In my opinion authors should find a reason why neat resin had higher particle size and attribute this to lower viscosity. Maybe, if authors did not analyzed product of PET glycolysis, and only used theoretical values of -OH and -COOH numbers, this could be a reason. For example if some of EG was lost this could derange calculations for resin feed composition.

Response: Yes, the viscosity of alkyd resin containing waste PET glycolysis products is significantly higher than that without PET. This can be attributed to three reasons. On the one hand, the shorter the oil length, the greater the viscosity. On the other hand, it is because the T_g of the alkyd resin based on waste PET contains terephthalic acid is higher than those of the alkyd synthesized from phthalic anhydride. Finally, in water-reducible resins, the smaller the particle size, the stronger the intermolecular forces, resulting in greater viscosity.

8. Page 7 line 56. Authors wrote " It can be expected that when the autoxidation water-borne alkyd resin film was cross-linked, it would have a larger number average molecular weight and mass average molecular weight, as well as better film properties." This is not something that "could be expected" but it is. After crosslinking M_n and M_w increases so high that it is no longer viable to talk about molecular weight.

Response: Yes, this sentence has been revised, please see the revised manuscript.

9. Conclusion sounds like short list of Results. Authors should add short explanations why did they get these results and add why resins with PET showed smaller particle

size which affected viscosity.

Response: Yes, conclusion sounds like short list of Results, conclusion had been rewritten, and short explanations had been given for get these results, please see “Conclusion” section in the revised manuscript.